# Microstructure Transformation in Laser Additive Manufactured NiTi Alloy with Quasi-In-Situ Compression

**DOI:** 10.3390/mi13101642

**Published:** 2022-09-30

**Authors:** Xiao Yang, Shuo Wang, Hengpei Pan, Congyi Zhang, Jieming Chen, Xinyao Zhang, Lingqing Gao

**Affiliations:** 1Luoyang Ship Material Research Institute, Luoyang 471023, China; 2Henan Key Laboratory of Technology and Application of Structural Materials for Ships and Marine Equipments, Luoyang 471023, China

**Keywords:** laser additive manufacturing, compression, microstructure transformation, dislocation pinning, recovery ability

## Abstract

For NiTi alloys, different additive manufacturing processes may have different compressive recovery capabilities. In particular, there are relatively few studies on the compressive recovery ability of NiTi alloys by the laser-directed energy deposition (LDED) process. In this paper, the compression recovery properties of NiTi alloys with the LDED process were investigated quasi-in-situ by means of transmission electron microscopy, an electron backscatter diffractometer, and focused ion beam–fixed-point sample preparation. The results showed that the material can be completely recovered under 4% deformation and the B19’ martensite phase content and dislocation density are basically unchanged. However, the recovery rate was only 90% and the unrecoverable strain was 0.86% at 8% deformation. Meanwhile, the B19’ martensite phase content and dislocation density of the material increased. Furthermore, with the increase in deformation, the relative dislocation pinning effect of the Ti_2_Ni precipitated phase in the alloy was enhanced, which reduced the compressive strain recovery to a certain extent.

## 1. Introduction

NiTi alloy, with a nearly equal atomic ratio, has the most excellent performance, which is the most common application among many known shape memory alloys. It has many advantages, such as superelasticity, the shape memory effect, fatigue resistance, a low elastic modulus, and biocompatibility [1,2,3,4]. As we know, the most widely used commercial methods for producing NiTi alloy parts are vacuum arc melting and vacuum induction melting, followed by hot working or cold working. It is likely that casting would lead to segregation defects [5]. Meanwhile, vacuum induction melting has the drawback of crucible contamination [6].However, when NiTi components undergo cold working, they encounter springback and make the overall dimension difficult to be shaped, hindering the application of nickel-titanium alloys.Additive manufacturing (AM), which has emerged in the past decade, is a new technology that can realize the integrated molding of complex components, and a lot of research results have been obtained [7,8,9,10]. Among them, selective laser melting (SLM) technology [11,12,13,14] and laser-directed energy deposition (LDED) technology [15,16,17] are typical metal additive manufacturing technologies, which could achievethe rapid formation of metal parts.

In the recent years, laser additive manufacturing has become an ideal preparation method for complex NiTi alloy components, and related technologies have developed rapidly, especially SLM technology [18,19,20,21].Dadbakhsh et al. [22] showed that SLM parameters have a great influence on the phase transition temperature and mechanical response of dense porous NiTi alloys. Meier et al. [23] studied the effect of deposition orientation on the compressive properties of SLM Ti-rich Ni_50.2_Ti_49.8_ (at.%) and found that crystal orientation had no significant effect on the compressive properties of these samples. Yang et al. [24] investigated the additivemanufacturing process of gradient NiTi alloys and obtained the gradient martensite phase by adjusting the process parameters. Bormann et al. [25] focused on the microstructure and texture of NiTi alloys fabricated by SLM and considered the effects of different processing parameters and the scanning rate. Haberland et al. [26] researched the superelasticity and cyclic response of NiTi alloys fabricated by SLM. Lu et al. [27] learned the simultaneous improvement of mechanical properties and shape memory properties by proper precipitation of the Ti_2_Niphase and its mechanism of action. It can be found that a lot of research has been carried out on NiTi alloys with SLM technology, from process parameters and structure to properties.

Compared toSLM technology, LDED technology has lower cost, faster formation, and the ability to form large components and has received extensive attention and research in the industry. Bimber et al. [28] studied the spatial anisotropy of NiTi alloys prepared by LDED and explored the differences of microstructures with different deposition heights. Wan et al. [29] studied the phase transformationbehavior, microstructure evolution, and elastocaloric properties of NiTi alloys prepared by LDED. It is confirmed that NiTi alloys prepared by the LDED process haveexcellent properties and arenot inferior to as-cast NiTi alloys. Research on the compressive properties of NiTi alloys prepared by AM technology has also been reported recently, but the research results are relatively few. Marattukalam [30] prepared NiTi alloys by means of LDED technology; 8% strain was recovered under a pre-compression of 10%, and the recovery rate was further improved after heat treatment. Andani et al. [31] fabricated and designed NiTi shape memory alloys with porous forms by selective laser melting, which exhibited a good shape memory effect, with a recoverable strain of about 5% and functional stability after eight cycles of compression. In addition, the stiffness and residual plastic strain of porous NiTi alloys were found to depend highly on the pore shape and the level of porosity. Moghaddam et al. [32] fabricated Ni-rich NiTi components by using the additive manufacturing process with 250 W laser power, 1250 mm/s scanning speed, 80 µm hatch spacing and without any post-process heat treatments. The results showed superelasticity with 5.62% strain recovery and a 98% recovery ratio.Chen et al. [33] successfully fabricated uniform and graded gyroid cellular structures of NiTi alloys by laser powder bed fusion additive manufacturing. The results showed that all laser powder bed fusion NiTi structures exhibited similar nominal compressive elastic moduli (5–7 GPa) ashuman bones. The resultsof Zhang et al. [34] showed that the energy input balances against the energy output during cyclic loading of porous NiTi alloys after performing several compression cycles as ‘training’ and the porous NiTi alloys exhibit reliable linear superelasticity and a stable elastic modulus, with strain as high as 4%.

Most of the research on NiTi alloys with AM technology has mainly focused on their functional properties and process parameters. However, there is little systematic in-depth study on the microstructure transformation of the material after compression, especially the microstructure transformation and recovery ability after compression under the LDED process. In this paper, an in-situ compression test was carried out on LDED NiTi samples, and the microstructure transformation and recovery ability of shape memory alloyswere researched in detail.

## 2. Materials and Methods

### 2.1. Material Preparation Process

The spherical near-equiatomic Ni_50_Ti_50_ alloy powder used in this experiment was purchased from China Shenzhen Micro-Nano Additive Technology Co., Ltd. The powder particle size range was 60–150 μm, and the particle size distribution was uniform and exhibited good sphericity. In this paper, LDED technology was used to realize the preparation of samples. The samples were fabricated in LDED equipment consisting of a 4 kW semiconductor laser, a five-axis numerical control working table, a coaxial powder feeder nozzle, and a chamber for circulating inert gas. The preparation process is shown in Figure 1 The samples were deposited using abidirectional scanning strategy, alaser power of 1800 W, a scanning speed of 600 mm/min, a powder feeding rate of 15 g/min, and a layer thickness of 0.5 mm. Before the LDED process, the mixed powders were dried in a vacuum oven for 2 h at 100 ± 5 °C to eliminate moisture absorption and ensure good flow ability. The forged NiTi plate, with a dimension of 150 × 50 × 10 mm^3^,was used as the substrate during the LDED process. Since the oxygen content would have a significant impact on the formation of NiTi alloys, the forming chamber was evacuated before fabrication and high-purity argon (>99.999%) was used to reduce the introduction of oxygen and to ensure that the oxygen content in the chamber was less than 100 ppm during the forming process.

During the LDED process, the NiTi alloy powder is exposed to a laser beam with high energy density. The powder is rapidly heated to a temperature above the melting point or even the boiling point. When the laser beam leaves, the melt solidifies quickly (depending on the LDED process parameters and materials). The previously solidified material undergoes a cyclic heating/cooling process, which could promote the continuous epitaxial growth of grain sand evolved to form columnar grain morphologies [35].

### 2.2. Samples and Test Preparation

The Φ10*20 mm mechanical compression samples were cut from the LDED sample by wire cutting, and to minimize the error of the compression test, the compressed upper and lower surfaces weresanded with sandpaper and polished with flannel to ensure sufficient smoothness and parallelism. Next, the quasi-in-situ [36,37] observation area (the light-blue box in Figure 2) of the sample was marked by an American Wilson VH3100 microhardness tester (made by Buehler Corporation of America) with aforce value of 50 gf, and the distance between the hardness points was 1.5 mm.

The samples with 0% deformation, 4% deformation, and 8% deformation were observed by electron backscatter diffraction (EBSD) and a transmission electron microscope (TEM) to achieve a comparative analysis of the microstructure and phasecomposition. EBSD analysis was performed with a medium-speed and high-resolution electron backscatter diffractometer (American EDAX Co., Ltd., Mahwah, NJ, USA), and TEM analysis was performed with a JEOL-2100 microscope (Japan electronics Co., Ltd., Tokyo, Japan).

EBSD samples needed to be prepared before analysis. As pecific sample preparation process was followed. First, the sample surface was polished to a particle size of 2.5 μm using silicon carbide grinding papers from 180 to 1200 grit. Second, the sample surface was subsequently electro-polished (electro-polishing refers to the process in which metal is subjected to special anodic treatment in a certain composition of polishing liquid to obtain a smooth and bright surface)—in a solution of HNO_3_:CH_3_OH (1:10 volume fraction) under 20 V for 15 s.

TEM samples were prepared by a dual-beam microscope system (American Thermo Electron Corporation) equipped with FIB and SEM at the same time, which can realize the precise preparation of TEM specimens in a fixed-point area. The details are as follows: First, a layer of Pt with a thickness of about 0.5 μm was deposited on the surface of the sample by an electron beam, which could avoid damage to the surface because the electron beam was small and the energy was low. Second, a 2 μm layer of Pt was deposited on the surface of a 0.5 μm Pt layer by ion beam deposition, which was used to eliminate the damage to the sample surface in the process of cutting and thinning in the later stage. Third, the area protected by Pt was processed by ion-beam-cutting technology, and then, the micro-nano-processed sample was transferred to as pecial copper mesh by a nano-manipulator. Lastly, the thickness of the sample was reduced to about 200 nm with a small current ion beam by the tilt function of the sample stage, and the surface deformation layer was also removed. Finally, the sample was transferred to the TEM for analysis.

## 3. Results and Discussion

### 3.1. Material Microstructure and Thermal Analysis

The morphology of the shape memory alloy prepared by the LDED method under anoptical microscope (OM; equipment model Leica DMI5000M, made by Leica Geosystems GmbH Vertrieb) is shown in Figure 3a. The sample was polished using silicon carbide grinding papers from 180 to 1200 grit and then polished with flannel to remove surface scratches and deformed layers. The result showed that the NiTi alloy prepared by LDED has fewer macro-defects, no microcrack defects, and a small number of pore defects. The small numbers of pore defects can be attributed to the unstable molten-pool-induced spatter [38,39] or the air flow introduced into the melt pool, but the overall forming quality was good. The microstructure of the alloy was dominated by columnar grains, and a few equiaxed grains were scattered in the interior. The transformation from columnar grains to equiaxed grains occurred. The lifting amount of each layer of the technology was high, and the diameter and expansion range of the laser spot were also large. The deposition size of the single layer of the single melting path was 1 order of magnitude larger than that of SLM technology. The spot diameter of the laser three-dimensional formationreachedseveral millimeters, which made the microstructure of the Ni_50.8_Ti_49.2_ alloy produced by laser additive change greatly in the temperature field (temperature gradient and direction) between different molten pools and less in a single molten pool. The grain hada larger expansion range between single cladding layers, so its grain size was larger. The growth morphology of grains depends on the ratio of temperature gradient (G) to solidification rate (R). The larger the G/R, the easier formation of columnar grains. The smaller the G/R, the easier the formation of equiaxed grains [40].

In NiTi alloys with a near-equiatomic ratio, the Ni/Ti atomic ratio significantly affects the phase transformation temperature and an increase inNi content sharply reduces the phase transformation temperature, ranging from 50 to 51 at.%, and there is a temperature change of about 100 K/at.% [41]. Some studies have shown that a high-energy input causes serious vaporization and ablation of Ni during deposition, resulting in a decrease inthe Ni content of the as-built sample matrix and an increase in the phase transformation temperature. Figure 3b shows the differential scanning calorimetry (DSC) curves of different positions of the sample. It can be seen that the start temperature of martensitic transformation (Ms) and the end temperature of reverse transformation (Af) of the sample were both lower than room temperature, so the main phase composition of the sample was the B2 parent phase at room temperature. Based on the fact that Ms and Af have great influence on superelasticity at room temperature, the phase transformation temperature indicates that the deformation of the sample at room temperature is superelasticity.

### 3.2. Phase Transition Behavior under Compressive Strain

The LDED sample was compressed on anMTS mechanical testing machine according to 4% and 8% deformation of the original size. The stress–strain curve of compression is shown in Figure 4. When the sample was at 4% deformation, the recovery rate was 100%. However, it had 10% plastic deformation, with 8% compression deformation. It is noteworthy that in the compression deformation at 4%, as shown in Figure 4a, the critical stress for stress-induced martensitic transformation was about 360 MPa and the critical strain was about 3% (determinedby the tangent method, and point E is the end point of 4% deformation). The difference in the critical strain is due to the compression of the sample through a smaller strain, which activates the deformation path of more martensitic variants [42], and stress-induced martensitic transformation is more likely to occur under a large strain.

When a load is applied, the parent phase (A) experiences elastic loads (A,B). At a specific loading level, the loading path intersects the onset surface of the martensitic transformation on the phase diagram, which marks the beginning of the transformation to martensite. It is worth mentioning that the stress-induced transformation from austenite to martensite is accompanied by the generation of a large amount of inelastic strain. In the transformation process (B,C), when the stress level reaches Mf, it means the end of the phase transformation. When the slope of the stress–strain curve changes significantly, the martensitic transformation is complete. When we continue to increase the force, only the elastic deformation (C,D) of the self-cooperative martensite occurs and there is no further phasetransformation. When the stress is gradually unloaded, the phase transformation causes the recovery of the strain. The end of the transformation to austenite is indicated by the point at which the stress–strain unloading curve re-enters the elastic region of austenite, and the material then undergoes elastic deformation back to A. The length of the blue line in Figure 4b is 0.86%,which is the amount of irreversible strain caused by plastic deformation.

### 3.3. Hardness Analysis after Different Compression

The microhardness test was carried out using American Wilson VH3100 (made by Buehler Corporation of America), with a force of 100 gf, and was completed within 10 min after compression. As shown in Figure 5, with the increase indeformation, the hardness value of the sample also increased, which indicates that the amount of deformation has a positive correlation with microhardness. In addition, the microhardness increased significantly at 8% deformation, reaching 16.9%. It is well known that the main mechanism of deformation strength is based on dislocation movement. With an increase in deformation in the LDED sample, the dislocation density of the sample increased, and then, the strength and hardness of the surface increased accordingly. At 4% deformation, the recovery rate of the sample was 100%. Although no obvious martensite phase was identified, the dislocation density still increased during the deformation process. In addition, at 8% strain, the deformation of the sample was not fully recovered, which significantly introduced plastic deformation. At the same time, as shown in Figure 4, the appearance of the martensite phase may have also promoted an increase in the microhardness value [43].

### 3.4. EBSD Analysis

Figure 6 shows the quasi-in-situ EBSD morphologies of the 0%, 4%, and 8% deformation.

The inverse pole figure orientation contrast maps of the original state, 4% deformation, and 8% deformation samples are shown in Figure 6a–c, respectively. The results showed that the 0% deformation sample was a single-parent phase with a B2 crystalstructure and hada strong texture of <100>‖BD (deposition direction). The sample with 4% deformation did not exhibit obvious martensite and dislocation defects, which are caused by deformation. Thisshows that the sample with 4% compression deformation completely recovered and there wasbasically no structural deformation and no obvious accumulation of dislocation density. However, when the amount of compression deformation increased to 8%, the content of the deformed martensite phase increased suddenly and reached 4.5%. Meanwhile, the kernel average misorientation (KAM) diagramsof the original state, 4% deformation, and 8% deformation samples are exhibited in Figure 6d–f, respectively. KAM is the most commonly used method in local misfit angle analysis. It is generally used to describe the local strain distribution of crystalline materials and is especially suitable for describing the strain distribution at the grain and phase boundaries of crystalline materials after deformation.It can be seen from the KAM diagram that as the amount of deformation increased, the color of the quasi-in-situ region of the material became darker. Therefore, it can be concluded that the stress distribution in the quasi-in situ region also increasedsignificantly when the deformation increasedto 8%.

The compressionwith 8% deformation caused a certain irreversible plastic deformation, and the stress-induced martensite was elongated and distributed at a certain angle (30–40°) to the deposition direction (BD). It is worth noting that with an increase in deformation, the strength of the preferred orientation slightly decreased. The decrease in the texture in the8% deformation sample may be related to the presence of the B19’ martensite phase, leading to adecrease in the B2 parent phase.

### 3.5. TEM Analysis

The quasi-in-situ TEM samples with different deformations were cut by an FIB. The phase morphology of the original sample is shown in Figure 7a. We concluded that there wasno obvious dislocation aggregation phenomenon in the original sample and there were two different phases in the original sample, which are marked as regions b and c. The selected area electron diffraction (SAED) of regions b and c is shown in Figure 7b and Figure 7c, respectively. Region b is the B2 parent phase. Region c was the Ti_2_Ni phase, which is the main precipitation phase of the material.

When the sample was subjected to 4% deformation, the microstructure of the material changed significantly. As shown in Figure 8, there was obvious dislocation aggregation in the quasi-in-situ area of the sample. Meanwhile, the phase of the material did not change significantly and the whole quasi-in-situ area of the sample was still mainly the B2 parent phase. However, it is worth noting that a small amount of the B19’ martensite phase had begun to exist in the individual positions of the quasi-in situ region. The morphology of dislocation aggregation and SAED of the B19’ martensite phase are shown in Figure 8.

The morphology of dislocation aggregation and SAED of the B19’ martensite phase with 8% deformation are shown in Figure 9. The results showed that the dislocation aggregation phenomenon in the quasi-in situ region was more obvious and the dislocation density increased significantly. Furthermore, there was more B19’ martensite phase in the whole quasi-in-situ region.

To further determine the relationship between the Ti_2_Ni precipitated phase and dislocation aggregation, we performed TEM analysis on the original sample, the 4% deformation sample, and the 8% deformation sample, and the results are shown in Figure 10. It can be seen that there were no obvious dislocations near the Ti_2_Ni precipitated phase of the original sample. However, when the sample was subjected to 4% deformation, there wasa certain degree of dislocation aggregation near the Ti_2_Ni precipitated phase. Moreover, the dislocation density around the Ti_2_Ni precipitated phase obviously increased; after 8% deformation, which means the hindering effect of the Ti_2_Ni precipitated phase on the dislocation was more obvious. The accumulation of dislocations leads to local stress concentration, so the stress-induced martensite is difficult to recover and the strain is retained.

## 4. Conclusions

The quasi-in situ analysis of LDED-prepared samples with 0%, 4%, and 8% deformation showed compression behavior can reflect the strength and plasticity of the sample, and it is a common method to characterize the mechanical properties of metal materials. According to the compression test, the microhardness of the material increases with an increase in the deformation amount, and the increase in microhardness can reach about 16.9% under 8% deformation. The increase in material strength is mainly due to the formation of deformed martensite and an increase in dislocation density.

Shape memory alloy materials have good superelastic properties, which can be fully recovered under 4% compressive strain, and the recovery rate can also be as high as 90% under 8% compressive strain.

In terms of microstructure transmission, under a compressive deformation of 4%, there is basically no B19’ martensite phase and the dislocation density does not change significantly. However, when the sample issubjected to 8% compressive deformation, the content of the B19’ martensite phase and the dislocation density in the material increasessignificantly.

With the increase incompressive strain, the dislocation density near the Ti_2_Ni phase increases and its dislocation pinning effect also increases. The dislocation pinning effect of the Ti_2_Ni precipitated phase reduces the deformation recovery performance of shape memory alloys toa certain extent.

## Figures and Tables

**Figure 1 micromachines-13-01642-f001:**
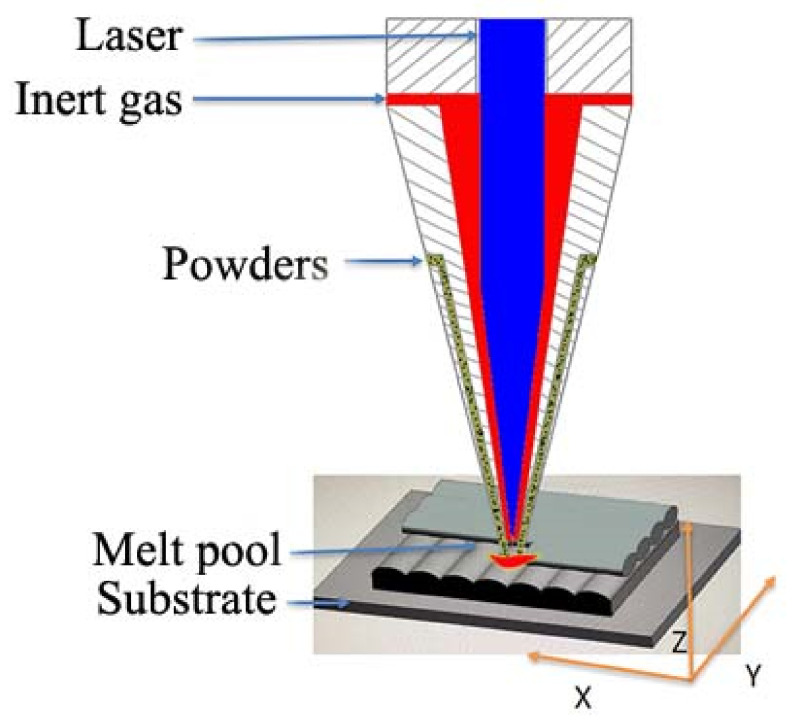
Basic structure diagram of LDED.

**Figure 2 micromachines-13-01642-f002:**
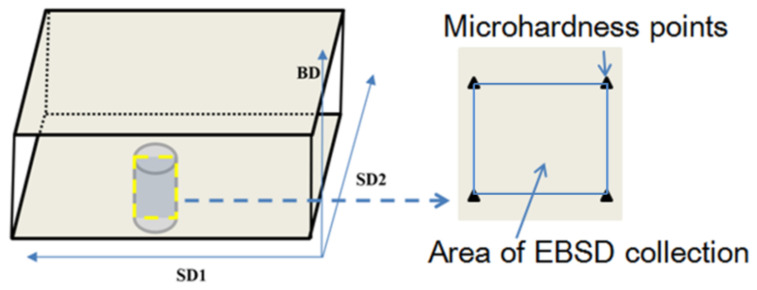
Schematic diagram of insitu region interception and preparation.

**Figure 3 micromachines-13-01642-f003:**
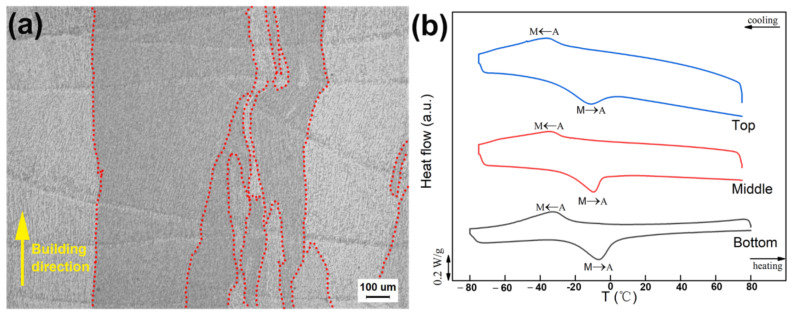
(**a**) Metallographic photos and (**b**) DSC curves of the bottom, middle, and top positions along the deposition direction.

**Figure 4 micromachines-13-01642-f004:**
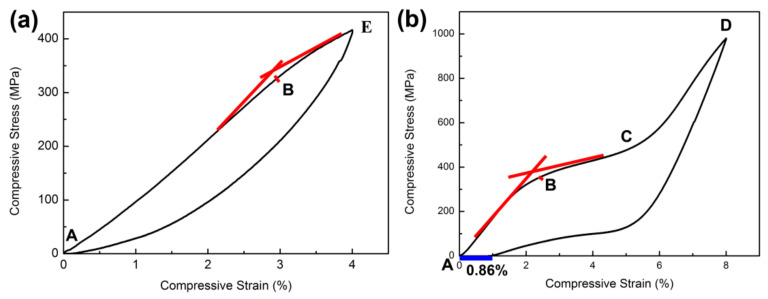
Stress–strain curve of compression: (**a**) 4% deformation and (**b**) 8% deformation.

**Figure 5 micromachines-13-01642-f005:**
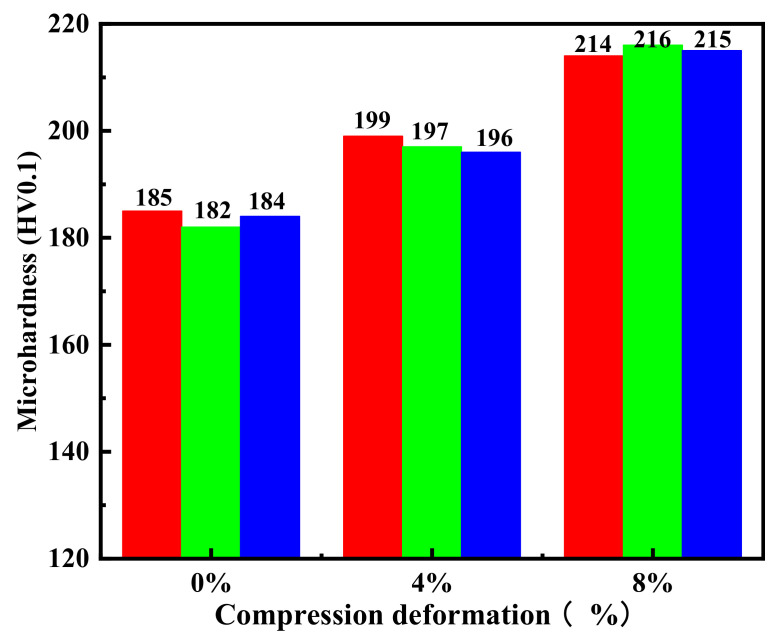
Microhardness values of samples at 0%, 4%, and 8% compressive deformation.

**Figure 6 micromachines-13-01642-f006:**
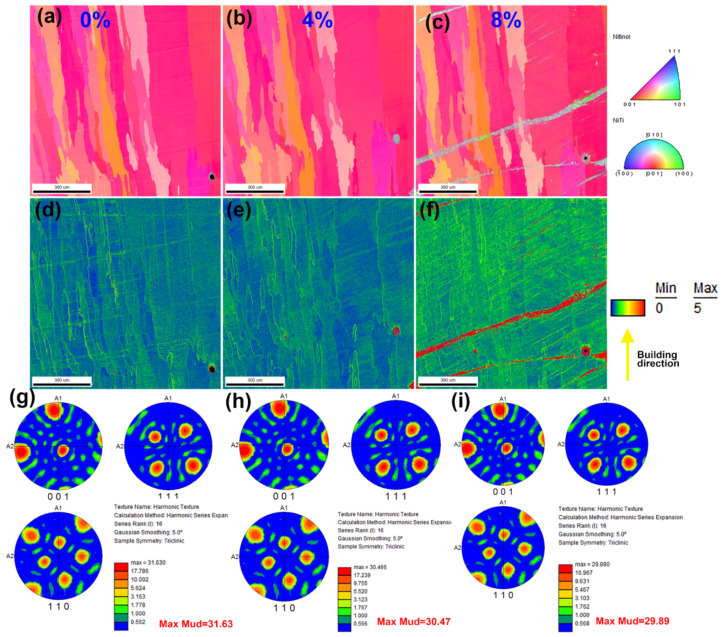
EBSD analysis of different samples: (**a**–**c**) inverse pole figure orientation contrast maps of the original state, 4% deformation, and 8% deformation samples, respectively; (**d**–**f**) KAM images of the original state, 4% deformation, and 8% deformation samples, respectively; and (**g**–**i**) pole figures of the 0% deformation, 4% deformation, and 8% deformation samples in {001}, {110}, and {111}, respectively.

**Figure 7 micromachines-13-01642-f007:**
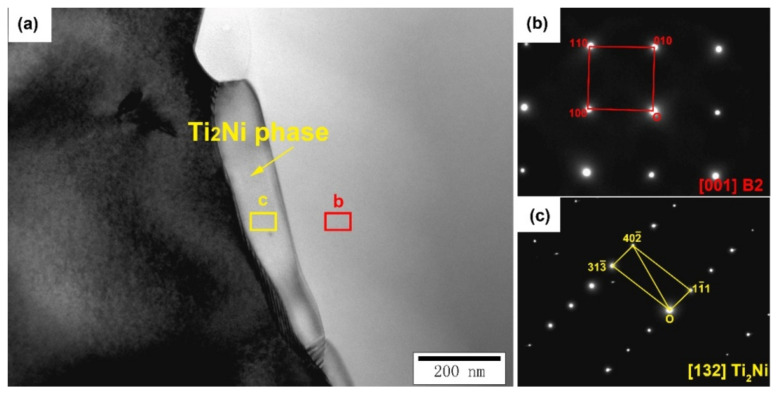
(**a**) is TEM morphologyanalysis of the 0% compressive deformation sample, (**b**) is SAED of the B2 parent phase (red box in (**a**)) and (**c**) is the Ti_2_Ni precipitatedphase (yellow box in (**a**)).

**Figure 8 micromachines-13-01642-f008:**
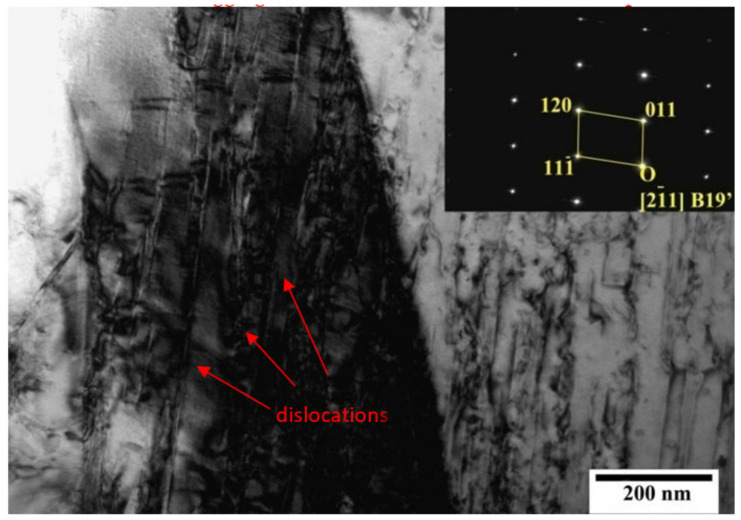
TEM analysis of the 4% compression sample and SAED of the B19’ phase.

**Figure 9 micromachines-13-01642-f009:**
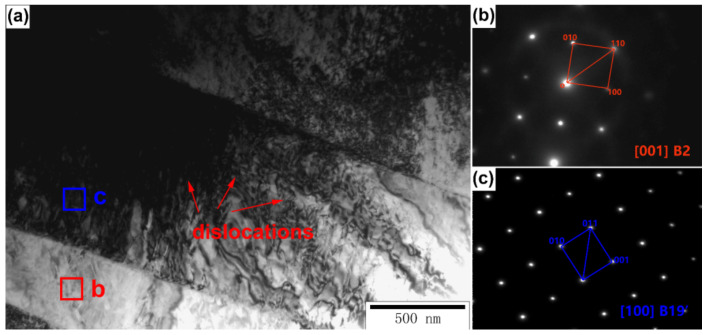
(**a**) is the TEM morphologyanalysis of the 8% deformation sample, (**b**) is the SAED of the B2 parent phase (red box in (**a**)) and (**c**) is the B19’ phase (blue box in (**a**)).

**Figure 10 micromachines-13-01642-f010:**
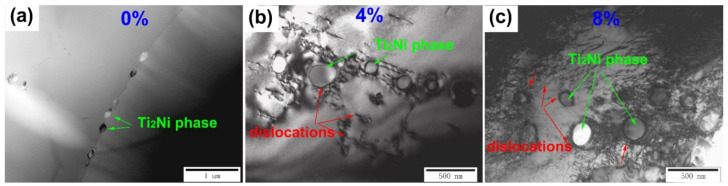
TEM bright-field images of (**a**) 0% compressive deformation, (**b**) 4% compressive deformation, and (**c**) 8% compressive deformation samples.

## Data Availability

The data presented in this study are available on request from the corresponding author. The data are not publicly available due to privacy.

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
