# Peer review of "Microstructure Transformation in Laser Additive Manufactured NiTi Alloy with Quasi-In-Situ Compression"

_micromachines, 2022, doi:10.3390/mi13101642_

Round 1

Reviewer 1 Report

The manuscript requires significant revision, as it lacks data that would allow me to assess the quality of the studies.

1) The LDED process should be described in more detail. What laser or laser machine did you use? There is no data on the Laser Power during the process, Scanning Speed, Spot Size, Track Displacement, etc. What did you use as a substrate (Fig. 1)? What size and configuration were the samples received?

2) The section on the equipment used is also poorly described.

3) In order to examine samples for specific properties - the “shape memory effect”, it is first necessary to conduct a study in order to show that the samples do not have defects, pores, cracks. It is necessary to carry out X-ray phase analysis of the initial samples (without deformation). It is possible to estimate the grain size. It is nessessary to compare with the literature data on the samples of this alloy, obtained in literature by laser cladding or smelting in a vacuum induction furnace.

4) Are there any data on the "shape memory effect" for literary samples obtained using additive technologies?

5) No data on martensitic transformation for a sample with 0% deformation.

6) It is impossible to estimate the dependence of the recovery rate on the deformation from just two points. A number of studies are needed for deformations from 4% to 10% to show how the recovery rate decreases depending on the degree of deformation and to what extent the results obtained meet the requirements for such alloys. Does it make sense to obtain such alloys by additive technologies?

Author Response

We thank the reviewers for their critical reading of the manuscript. The suggestions are helpful for us to improve the manuscript. Our responses to the reviewers’ comments are listed in the attachment and all the modifications are highlighted using red font in the revised manuscript.

Reviewer 2 Report

In this article, the quasi in-situ compression test is carried out on the LDED NiTi samples, and the microstructure transformation of shape memory alloy was researched in detail.

(1) L52-53 "The powder particle size range is 60~150μm and particle size distribution is uniform and exhibits good sphericity." : How did you checked the size of the particle size and sphericity?

(2) Figure 3 : write abstract about (a) and (b) definitely.

(3)  In figure 3 (a), at point B Strain 3.8%, Stress 400 MPa, however,  in figure 3 (b), at point B,  2%, 340 MPa. Explain why this difference is born.

Author Response

(The authors gave the same response as above.)

Round 2

Reviewer 1 Report

The manuscript can be accepted for publication after minor revision.

 1) You indicate that the TiNi alloy served as the substrate during the LDED process. Was the final test specimen Ti50Ni50 remove/cut from the substrate?

2) Fig. 2. It seems to me that it is necessary to additionally provide an optical photo of the microstructure after etching (the shape and size of the grains will be immediately visible). Then Fig. 2 will fully reflect the microstructure of the sample before testing.
